# Antireflection Structures for VIS and NIR on Arbitrarily Shaped Fused Silica Substrates with Colloidal Polystyrene Nanosphere Lithography

**DOI:** 10.3390/mi14061204

**Published:** 2023-06-07

**Authors:** David Schmelz, Guobin Jia, Thomas Käsebier, Jonathan Plentz, Uwe Detlef Zeitner

**Affiliations:** 1Institute of Applied Physics, Abbe Center of Photonics, Friedrich Schiller University Jena, 07743 Jena, Germany; 2Leibniz Institute of Photonic Technology (Leibniz-IPHT), 07745 Jena, Germany; 3Fraunhofer Institute for Applied Optics and Precision Engineering IOF, 07745 Jena, Germany; 4Department of Applied Sciences and Mechatronics, Munich University of Applied Sciences, 80335 Munich, Germany

**Keywords:** antireflection, moth-eye nanostructures, colloidal lithography, polystyrene nanospheres, Langmuir-Blodgett, reactive ion etching, fused silica lens

## Abstract

Antireflective (AR) nanostructures offer an effective, broadband alternative to conventional AR coatings that could be used even under extreme conditions. In this publication, a possible fabrication process based on colloidal polystyrene (PS) nanosphere lithography for the fabrication of such AR structures on arbitrarily shaped fused silica substrates is presented and evaluated. Special emphasis is placed on the involved manufacturing steps in order to be able to produce tailored and effective structures. An improved Langmuir-Blodgett self-assembly lithography technique enabled the deposition of 200 nm PS spheres on curved surfaces, independent of shape or material-specific characteristics such as hydrophobicity. The AR structures were fabricated on planar fused silica wafers and aspherical planoconvex lenses. Broadband AR structures with losses (reflection + transmissive scattering) of <1% per surface in the spectral range of 750–2000 nm were produced. At the best performance level, losses were less than 0.5%, which corresponds to an improvement factor of 6.7 compared to unstructured reference substrates.

## 1. Introduction

In the fields of optics and photonics, increasing diversification and specialization with regard to the design and materials of optical systems is a current trend. The development of material- and form-specific solutions benefits from this. A typical case here is the antireflection (AR) effect of particular optical components. For special shapes such as free-form [1,2,3,4] or structured [5,6,7] surfaces, extended spectral ranges, or application in environments with extreme conditions [8,9], conventional AR coatings are often not readily applicable. AR structures, also called moth-eye structures, are a possible alternative for this purpose [10,11]. These are conical subwavelength structures that cause a continuous increase in the effective refractive index due to the continuous increase in the fill factor at the transition from air to the optical device. Reflections based on index jumps are thereby prevented or reduced. The advantages of these structures are their large spectral bandwidth and the material homogeneity. Moreover, they can be combined with coatings for more sophisticated optical solutions [12,13,14].

AR structures can be differentiated by their arrangement: Deterministic structures [15,16,17,18] offer good regularity and control of the structures. However, they are very expensive due to the lithography or direct writing effort required and are difficult to apply on non-planar surfaces. Stochastic structures [1,5,19,20,21,22], on the other hand, are inexpensive and more flexible in terms of surface shape. However, they are difficult to control due to the statistical variances regarding their dimensions and due to the process parameter-sensitive fabrication. Their irregular structure produces optical scattering [23] and inevitably requires compromises in terms of optical performance. This prevents specifically customized solutions for specific optical devices. In-between are structures produced with lithographic self-assembly methods [3,23,24,25]. Due to their regularity, they allow for a good adaptation to the optical requirements. On the other hand, they can be applied in a relatively cost-effective and flexible way on arbitrarily shaped substrates.

In this paper, the application of an improved Langmuir-Blodgett (LB) self-assembly lithography technique is presented. In this process, polystyrene (PS) nanospheres with a selected diameter are deposited on a substrate in a self-arranging monolayer. This creates a hexagonal array that can be used as a pattern for further lithographic processes. The pattern is first transferred into an underlaying Cr layer that subsequently serves as mask for etching into the fused silica substrate. The produced AR structures are afterwards optically evaluated with regard to their AR effect.

Colloidal lithography techniques are often used in combination with metal-assisted chemical etching (MACE) for the fabrication of silicon structures. Typically, PS nanospheres serve as negative masking for the deposition of a noble metal that is used for the etching process [26,27,28,29]. MACE is mainly applied for patterning silicon and cannot be transferred to glass materials. In this paper, PS nanospheres serve as a positive mask [9,23,30,31,32], and reactive ion etching (RIE) techniques are used for the creation of the structures. This approach is much more broadly applicable on different materials and shapes. Using this method for the generation of AR structures on fused silica substrates [9,13,23,33] comes with certain challenges regarding the fabrication of suitable structures. The insertion of an additional Cr layer [15] between the PS spheres and the substrate increases the etch selectivity and thus allows for better process adaptation. Consequently, sophisticated structures with a high aspect ratio, smooth sidewalls, and an adapted structure profile can be fabricated that then achieve better AR performance. This paper aims to present a guideline for the fabrication of such AR structures on fused silica substrates. It focuses on certain fabrication aspects that should be considered in order to obtain structures that are most suitable for reaching a high transmission level.

## 2. Fabrication

Figure 1 illustrates the steps of fabrication of the AR nanostructures. The process starts from a fused silica substrate with a 30 nm thick Cr layer deposited by ion beam deposition (IBD). The used substrates were IR-grade fused silica wafers of 1 mm thickness and 50.8 mm (2″) diameter (see Figure 2), and planoconvex aspherical lenses of 2 mm thickness, 25.0 mm diameter, and 46.07 mm radius of curvature in Corning 7980 UV-grade fused silica glass. Figure 2 shows a single-side patterned 2-inch wafer in comparison to a double-side-polished (dsp) reference wafer. There are scratches from handling during the investigations on the left and bottom edges of the sample as well as some inhomogeneities.

Regarding the fabrication, the focus is on four manufacturing steps:**1.** **Deposition of PS nanospheres:** A compact PS monolayer is formed by a self-organized arrangement of PS spheres on the water surface and subsequent compression with surfactant addition. A slow drainage of the water enables their deposition on a substrate located at the bottom of the water basin.**2.** **Shrinkage of PS nanospheres:** An O_2_ plasma RIE step ensures the shrinkage of the individual PS nanospheres and thus the adaptation of the masking to the subsequent structuring process.**3.** **Transfer into Cr mask:** The PS masking is transferred into the underlaying Cr layer using a chlorine-based RIE process. A key factor here is the generation of rounded edges.**4.** **Etching of AR structures:** The Cr mask is etched into the substrate material by a stretched proportional transfer with inductively coupled plasma reactive ion etching (ICP-RIE). This forms vertically tapered AR structures in the substrate.

The method for preparing the PS monolayer is an improved version of the Langmuir-Blodgett (LB) method [26]. Instead of moving mechanical barriers, like in the standard LB process, the compression of the PS spheres is induced by surfactant functionalization. 100 µL of the original suspension is diluted with 1100 µL of ethanol. Subsequently, 5 µL of hexylamine is added for the surface functionalization of the PS nanospheres. The mixture is then sonicated for 30 min. The sample is immersed beneath the water of a Petri dish, and 100 µL of the prepared suspension is slowly injected into the water surface by a syringe. The PS nanospheres spread over the water surface and form a loose monolayer. Then, ca. 50 µL of a 10 wt% sodium dodecyl sulfate (SDS) water solution was injected with another syringe at the edge of the Petri dish. The SDS molecules form a monolayer and dynamically self-assemble themselves during the deposition, so that the pressure at the edge of the PS nanospheres is maintained. The loose PS sphere monolayer is compressed by the SDS and aligned in a hexagonal pattern by the effect of Van der Waals forces. A slow sinking of the water level enables their deposition on the underlaying sample (see Figure 3).

The developed process is significantly simpler, faster, and more fault-tolerant than the standard LB method. In such a way, a conformal deposition of the PS or other nanomaterials such as graphene can be realized on complex 3D shapes and is independent of the surface conditions of the substrate, such as hydrophilicity [34,35]. Moreover, the process offers a good potential for upscaling to larger areas.

The fabrication process was initially developed for structures with a 600 nm period [26], where an aqueous PS nanosphere suspension from Microparticles GmbH was diluted in ethanol and functionalized with hexylamine in an optimized ratio of PS:ethanol:hexylamine = 100:100:5 (in *v*:*v*). For applications in the VIS and NIR spectral ranges, the 200 nm period and the associated change to nanospheres with 200 nm diameter were modified, so that the ratio of the individual components needed to be optimized. In this work, the ratio of PS:ethanol:hexylamine = 100:1100:5 (in *v*:*v*) was the optimal condition.

The generated monolayer on the sample serves as a template for transfer into an underlaying 30 nm thick Cr layer. The period of the hexagonal pattern is defined by the diameter of the nanospheres. After the deposition, an O_2_ plasma etching step is performed in a RIE plasma etcher SI 591 from SENTECH Instruments GmbH. This shrinks the spheres into a desired size while they remain in their respective positions, and the period stays fixed (see Figure 4). This creates a sufficiently large etch aperture through which the subsequent etching process can attack and form separated structures. If this step is not performed, undesirable contiguous structures result, as shown in Figure 4d. It is important to optimize the amount of shrinkage, so that, on the one hand, the structures are well-separated from each other, and, on the other hand, sufficient etch masking material remains. This can be controlled by selecting the appropriate etching time. For the O_2_-RIE process, a standard recipe for the etching of organic photoresists with an RF power of 50 W and an O_2_ gas flow of 50 sccm was chosen. The shrinkage of the organic PS nanospheres is time-dependent. Figure 5 shows the measurements of the diameter of the shrunk PS spheres inspected with scanning electron microscopy (SEM). It shows a linear dependence between etching time and diameter. Variations in terms of shrinkage were observed for different sample sizes, which must be considered when setting the etching time.

For the transfer of the mask into the Cr layer, the choice of the etching technology is decisive. The applied chlorine RIE process (2 min, Cl_2_ 50 sccm, O_2_ 10 sccm, RF power 100 W) with its isotropic etching component shapes the Cr mask so that it has a slightly descending, rounded sidewall on the outside. This forms the basic shape for the structure profile that can subsequently be transferred into the underlaying fused silica substrate by a stretched proportional transfer. If an ion beam etching (IBE) process with only a physical etching component were used for the transfer instead, the resulting redeposits would produce mask profiles with a slightly outwardly increasing shape. These mask profiles would not be suitable for the following ICP-RIE process. This issue is illustrated in Figure 6, showing PS spheres with 600 nm period on a 30 nm thick Cr layer deposited on a silicon substrate, as silicon is clearly easier to image in an electron microscope than fused silica is. Of the two samples, one was etched with RIE and the other with IBE. The differences in the resulting shape of the Cr masks on the two samples are clearly visible.

The etching of the Cr mask is followed by an ICP-RIE etching step with CHF_3_ as the etchant gas at a gas flow of 12 sccm. The process is performed in an ICP-RIE plasma etcher SI 500C etcher by SENTECH Instruments GmbH at relatively high temperatures of 75 °C and a moderate ICP power of 180 W. The process was bias-controlled with a bias voltage of −180 V in the RIE chamber and 20 min duration. It can be considered a stretched proportional transfer of the Cr mask into the SiO_2_ substrate. Conical structures were formed in the fused silica 2-inch wafers, as shown in Figure 7a,b. After 20 min, the structures show positive sidewall slopes. In the bottom region, they are connected to each other. Further etching progress is stopped due to the RIE lag respectively aspect ratio dependent etching (ARDE) [36]. The resulting structures have a height of about 600 to 700 nm. With respect to the 200 nm period, this corresponds to an aspect ratio of 3 to 3.5.

The fabrication process is relatively sensitive towards the substrate material and shape. Transferring the process towards the planoconvex SiO_2_ lenses requires some adjustments of the process times in the O_2_-RIE and ICP-RIE steps 2 and 4. The O_2_-RIE process time was extended from 1 min to 1.5 min, as the etch rate of the shrinkage was lower. The ICP-RIE process time was shortened from 20 min to 15 min. This also resulted in slight changes of the structural profile. The structures on the lens are shown in Figure 7c,d. Instead of a conical profile with a constant sidewall angle, a rather bottle-like profile was formed that was close to the Klopfenstein profile [37]. Compared to other structural profiles optimized for the optimal coupling of light, the Klopfenstein profile is very effective in light coupling at comparatively low structural heights [38]. Hence, the structures are very suitable for effective antireflection.

After the etching of the SiO_2_ moth-eye structures, the remaining residuals of the PS spheres and the Cr layer are removed with O_2_-RIE and Cl-RIE. The fabrication process is then completed. The samples were inspected at different positions. They showed similar structural profiles over the entire sample.

## 3. Characterization

For the evaluation of the AR properties, the realized structures were characterized using a Lambda 950 spectrometer (Perkin Elmer) with an internal 150 mm diameter integration sphere. The structured samples were measured with respect to their specular transmission over a spectrum of 320–2000 nm and compared to unstructured reference samples. The measurement spot on the sample had a size of about 8 × 12 mm^2^. Besides the specular transmittance, the integration sphere enables the measurement of the total transmittance (consisting of specular and scattered transmissive light). This allows for the differentiation between losses via reflection and transmissive scattering. The former gives an estimate of the AR effect. The latter, on the other hand, is caused by defects in the structure (see Figure 7b). These defects mainly emerge during the deposition process of the PS spheres. In the case of a perfect defect-free assembly of the spheres, the AR structures would be in the subwavelength range. The defects, however, disrupt this subwavelength effect by adding lower spatial frequencies to the pattern, which causes scattering.

Figure 8 shows the measurement results of the structured 2-inch wafer sample (displayed in Figure 7a,b). It plots the specular transmission T_spec_ and total transmission T_total_ of the AR structured sample compared to a double-side polished (dsp) reference SiO_2_ sample. The reference sample measurements agree with the theoretical Fresnel reflectance values. From transmittance T of the reference sample, the transmittance of a single polished surface is calculated that is, in the case of a one-side structured sample, equivalent to an ideal AR effect. For an easier assessment of the AR structures, the transmittance of only the single-patterned surface is also determined and is plotted in Figure 8b.

The fabricated structures have a rounded rather than pointed profile towards the tip. This is beneficial for longer wavelengths where the structural height is lower in relation to the wavelength [23]. The reflectance curves show a broadband AR effect over the entire measurement spectrum. The specular transmission is significantly increased in the range of 500–2000 nm. Between 750 and 2000 nm, the specular transmittance per surface was above 99%. At its maximum at around 1100 nm, losses are less than 0.5%, which corresponds to an improvement factor of 6.7 compared to the polished reference SiO_2_ surface.

From 2000 nm towards shorter wavelengths, the transmittance slightly increases and reaches its maximum as the ratio between height and wavelength also increases. Ji et al. [23] describe this part of the spectrum as the range where the shape of the structures is most decisive for the AR effect. Left of the maximum, towards lower wavelengths, the arrangement of the AR structures is the most important factor. For wavelengths above 1000 nm, the transmissive scattering losses are rather neglectable. Towards shorter wavelengths, the gap between T_spec_ and T_total_ grows, indicating an increasing influence of the pattern defects that cause scattering. The decrease in specular reflectance already starts at relatively long wavelengths compared to the structure width of 200 nm. This can be explained by the randomization of the structure due to the presence of defects. Stochastically distributed AR structures exhibit scattering effects even for wavelengths that are significantly longer than the average structure width [23,39]. However, total reflectance T_total_ decreases as well, which indicates an increasing reflectance. This can be explained by the presence of larger areas with low inclination due to defects and the absence of steep structures in these areas. The shorter the wavelengths, the better even small areas are “resolved” by the incident light [40].

The measured values could be reproduced in several measurement series at different positions of the sample, which corresponds to the observations of a homogeneous structural morphology in the SEM images.

In addition to characterizing the planar substrates, which are well-suited for evaluating optical properties, there were also attempts to measure the structured lens sample. The difficulty here is that the refractive behavior of the lens strongly influences the optical beam path in the spectrometer. To reduce this influence, an aperture of 6 mm diameter was used. The aperture blocks the light that is further away from the optical axis, thus not illuminating the areas where strong refraction by the lens occurs. The aperture and the lens sample were placed at a moderate distance in front of the integration sphere. The problem here was that only the central region of the structured lens could be examined. In that area, structuring effects were already macroscopically visible, which influenced the results. Reducing the size of the measurement field results in noisier and less accurate measurement signals. The measured results are shown in Figure 9a. The transmission dip at around 1.4 µm wavelength is caused by OH^−^ absorption since the used material is UV-grade fused silica in contrast to the IR-grade wafer substrate. Stronger noise is also clearly visible. Apart from that, the values of the reference samples are in a similar range. While the AR effect in the upper wavelength range is still comparable to that of the wafer sample, a drop of the transmittance due to the defects on the lens sample can be seen already at higher wavelengths. Nevertheless, the AR effect based on the AR nanostructures could be principally demonstrated here as well.

This becomes even clearer in the second attempt that was performed to investigate the AR properties of the patterned lens sample. Here, the 6 mm aperture was omitted. The lens sample was attached directly in front of the integrating sphere using adhesive tape. In this attempt, it is difficult to control the orientation of the lens, which causes deflections in the optical path. However, the measurement spot is larger, which allows for a more general estimation of the AR effect of the lens sample. Furthermore, by placing the lens directly in front of the integration sphere, the transmissive scattered light can also be detected. The measured results are shown in Figure 9b. They show an obvious AR effect over the entire measurement spectrum. This confirms the assumption that, towards shorter wavelengths, losses are mainly caused by transmissive scattering instead of increasing reflection. In both attempts to evaluate the patterned lens, a strong improvement in transmission due to a significant AR effect was observed.

## 4. Conclusions

In this paper, a readily suitable method for the fabrication of antireflective (AR) moth-eye nanostructures on fused silica was presented. For this purpose, a lithographic method based on self-assembling colloidal polystyrene (PS) nanospheres was applied. An improved Langmuir-Blodgett (LB) process allowed for the nanospheres to be deposited on curved and free-formed substrates. This was demonstrated by the fabrication of AR structures on aspherical lenses of fused silica. In principle, this is applicable to arbitrary materials. Hence, other materials and spectral ranges are worthwhile subjects for future research. An established deposition process gives the opportunities of uncomplicated and prompt processing and modification of optical and photonic components, as no mask layout and no lithographic exposure is required. Apart from classical optical components such as lenses, it may be applied to photonic sensors, modulators, or integrated circuits based on silicon platforms. Conceivable examples are its application on monolithically integrated microlenses [41] or locally backside-thinned devices [42]. On the other hand, there are limitations when the surface area of the substrate to be coated is significantly larger than the area projected onto the water surface.

The insertion of an intermediate Cr layer of a few tens of nanometers significantly expands the design latitude of the desired structures. After transferring the PS pattern, the Cr layer served as masking for the etching of the AR structures. Its high selectivity towards glass in reactive ion etching processes enables the fabrication of structures with high aspect ratios for various materials. For even higher aspect ratios or materials with low etch rates, it might be suitable to apply a multiplexed instead of a static etching process.

Regarding the AR effect of the fabricated nanostructures, a significant suppression of reflection could be demonstrated. However, for the application of transmissive AR structures on glasses, the reduction of transmissive scattering is also essential. The scattering is mainly caused by patterning effects that particularly occur during the deposition of PS nanospheres. Further development of the presented improved LB process towards a defect-free deposition is, therefore, the key factor for the application at shorter wavelengths. Particularly important for this is a homogeneous nanosphere distribution. The optimized composition and high purity of the chemicals prevent the formation of surfactant residues that interfere with perfect compression from the outside. There are ideas and preliminary tests to convert the sequential directed self-assembly process into a continuous process and, ideally, use a roll-to-roll process. In this way, the process parameters can be adjusted very well, and the result can be monitored in situ. In addition to moving towards a defect-free deposition, the deposition on larger substrates might also be of interest for future research, as it enables the integration in common 100 mm and 150 mm fabrication lines.

## Figures and Tables

**Figure 1 micromachines-14-01204-f001:**
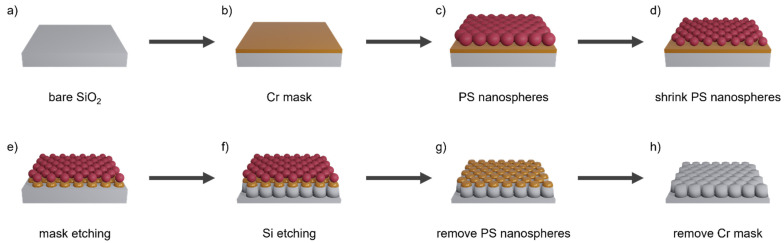
Schematic illustration of the fabrication process of AR nanostructures.

**Figure 2 micromachines-14-01204-f002:**
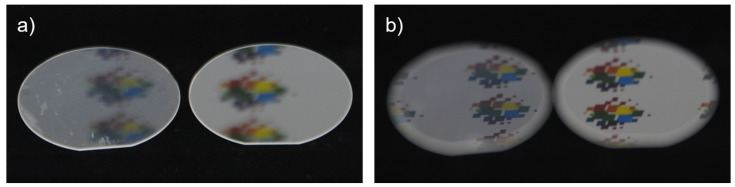
Photographs of two fused silica 2-inch wafers for comparison. A single-side patterned sample with fabricated AR nanostructures (left) is placed next to a double-side polished (dsp) reference wafer (right). The focus of the camera is set once (**a**) on the samples and once (**b**) on the reflected objects.

**Figure 3 micromachines-14-01204-f003:**
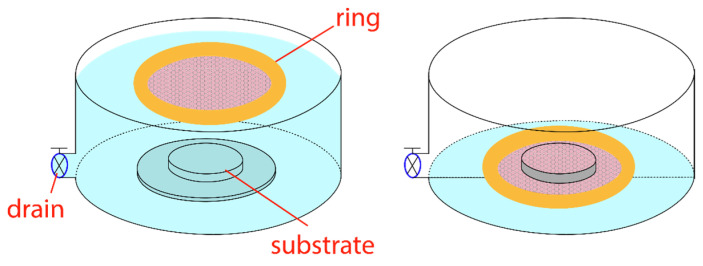
Principle of the monolayer deposition process. Nanospheres are compressed by adding surfactants and deposited on the substrate located at the bottom by draining the water with the aid of a guard ring.

**Figure 4 micromachines-14-01204-f004:**
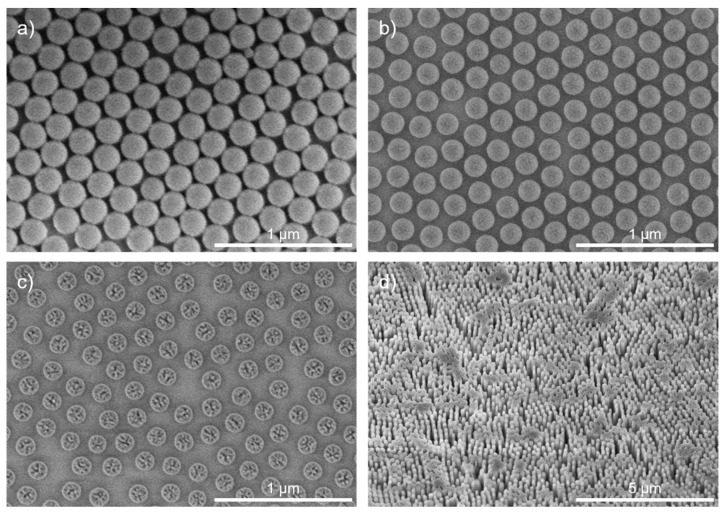
(**a**) PS nanospheres deposited on the surface of a Cr-coated SiO_2_ substrate without O_2_ plasma treatment. (**b**) PS nanospheres after O_2_ plasma treatment with a suitable amount of shrinkage. (**c**) Excessive shrinkage of nanospheres after overlong etching treatment. (**d**) Non-separated structures that emerge when the necessary O_2_ plasma step to shrink the PS spheres is omitted. The SEM images were taken (**a**–**c**) in top view and (**d**) at a 30° tilt angle.

**Figure 5 micromachines-14-01204-f005:**
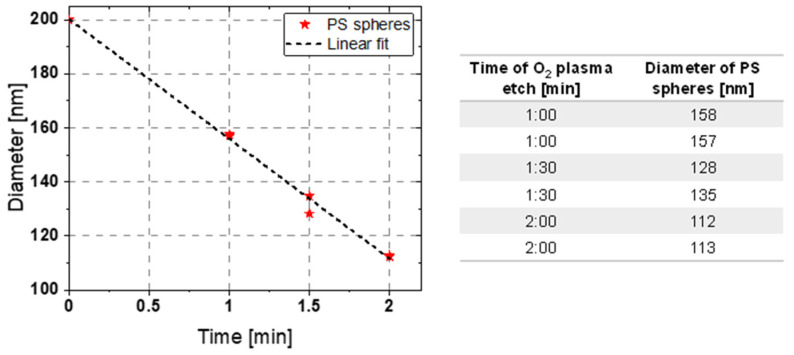
PS shrinkage after O_2_ plasma etching step with respect to the process time on different 2-inch wafer samples.

**Figure 6 micromachines-14-01204-f006:**
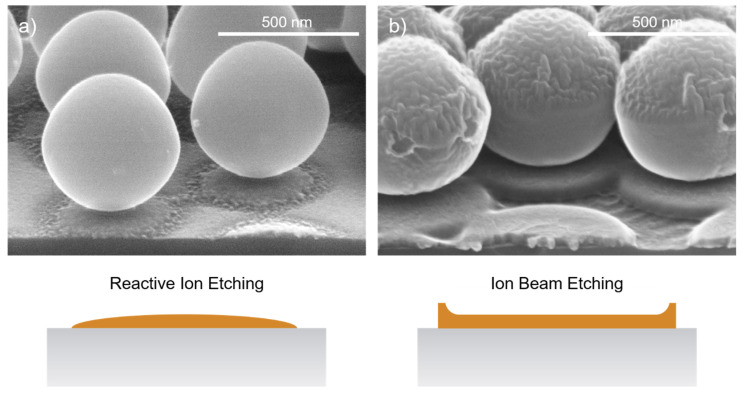
Comparison between (**a**) RIE and (**b**) IBE etching for opening the Cr mask layer. The isotropic component of the RIE etching shapes a rounded mask profile, while the IBE etching creates redeposits resulting in an elevation of the edge region. PS nanospheres with a diameter of 600 nm were used. The process was applied to Si substrates for better visualization in the electron microscope.

**Figure 7 micromachines-14-01204-f007:**
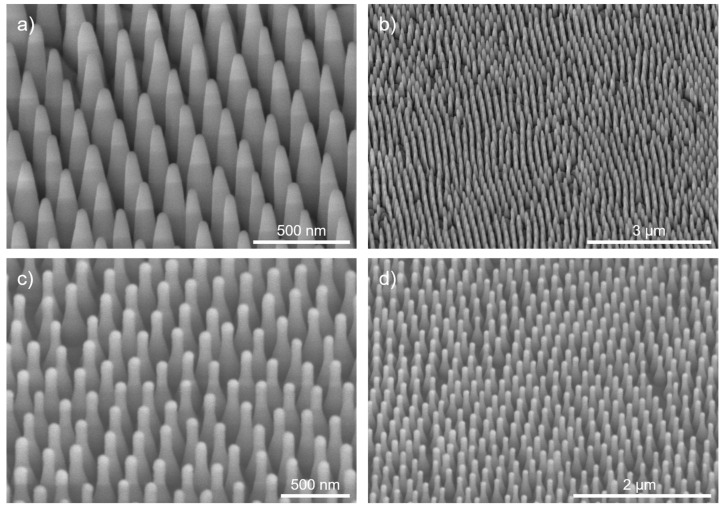
SEM images of fabricated SiO_2_ structures after ICP-RIE etching on (**a**,**b**) a plane wafer substrate and (**c**,**d**) on a planoconvex lens at 30° oblique view.

**Figure 8 micromachines-14-01204-f008:**
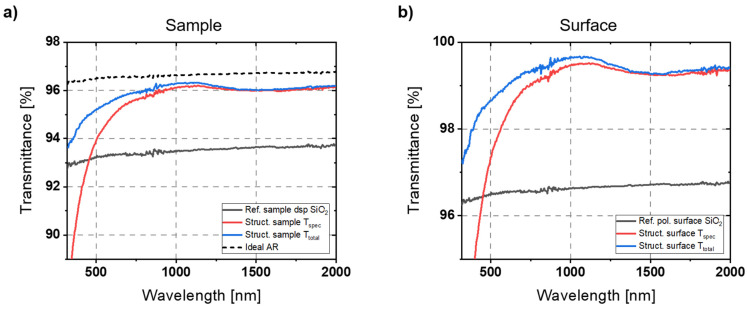
Transmission measurement of one-side structured 2-inch wafer sample and dsp reference sample. (**a**) Specular transmittance T_spec_ is compared to the total transmittance T_total_ (consisting of specular and scattered transmissive light) of the sample. (**b**) For the easier evaluation of the AR effect, the transmittance of only the structured surface is calculated and compared to the polished reference surface.

**Figure 9 micromachines-14-01204-f009:**
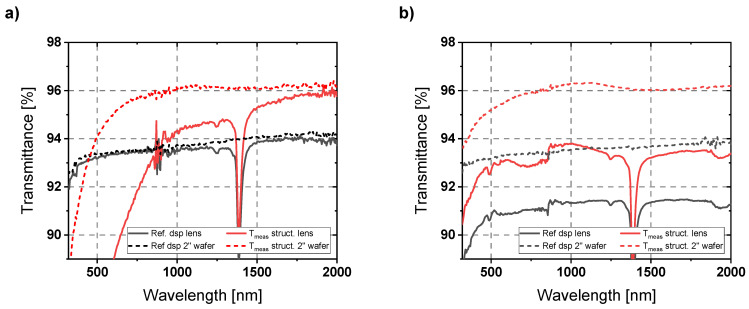
Transmission measurements of the AR structured lens sample compared to an unstructured dsp reference lens. (**a**) The lens was aligned in the center of the optical axis at a moderate distance from the integrating sphere of the spectrometer. An aperture with 6 mm diameter was placed just in front of the lens to cut out stronger refracted light. (**b**) The lens was attached directly in front of the integrating sphere, allowing for the transmissive scattered light to be measured. Control over the alignment of the lens is more difficult in this case, causing deflections in the optical path.

## Data Availability

The data that support the findings of this study are available upon reasonable request from the authors.

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
