# Peer review of "Antireflection Structures for VIS and NIR on Arbitrarily Shaped Fused Silica Substrates with Colloidal Polystyrene Nanosphere Lithography"

_micromachines, 2023, doi:10.3390/mi14061204_

Round 1

Reviewer 1 Report

Thank you for inviting me to review this manuscript. In regard to the presented manuscript, I appreciate the efforts made in developing a novel method for preparing anti-reflection (AR) structures on mirror surfaces using colloid polystyrene (PS) nanosphere lithography. The experimental results show that the AR structures produced have a total loss (reflection + transmissive scattering) of less than 1% in the spectral range of 750nm-2000nm (optimal result less than 0.5%), which is 6.7 times better than the untreated substrate. This study has significant potential for practical applications in areas where conventional AR coatings are not readily applicable.

However, I suggest that the authors improve the structure of the article by emphasizing the main points and contributions. Additionally, the number of references cited should be increased and recent studies related to AR coatings should be included.

Overall, this manuscript presents an interesting and valuable contribution to the field of AR structure and I recommend it for publication after minor revision.

1.Although the abstract describes the main content of the article, it is too detailed and does not highlight the focus and contribution of the article. You need to simplify and emphasize the innovation and importance of the article, so that the reader can quickly understand the core content of the article.

2. The conclusion is too long and incomplete, but rather a summary of the methods and experimental results described in the article. Further emphasis should be placed on the contribution, innovation and future research direction of the paper. Future research directions include further improvement of LB self-assembly technology to reduce deposition defects and exploration of AR structure applications in the field of optoelectronics. In addition, the preparation of AR structure on other materials can also be studied to expand its application range.

3. The number of citations needs to be increased, and it is suggested to increase the citations of literatures related to antireflective structure in recent 5 years. Most of the literatures cited are old studies ten years ago, which cannot accurately reflect the latest progress of specific material structures and related technologies.

Minor editing of English language required.

Author Response

Dear Sir or Madam,

Thank you very much for your thorough review of our manuscript. I think with your remarks we could definitely improve it. We have made several modifications which I will explain in the next lines by following your list of points.

  1. We have shortened the abstract by removing the kind of introduction sentences and, hence, directly getting to the point. Also, we removed some less important sentences in the middle part, thus, highlighting the core content.
  2. I agree with your assessment on our original conclusion. It was rather a summary of the content than a conclusion. Therefore, we modified the whole conclusion part. We removed several summarizing parts and highlighted the main contributional points of our work. Also, we wrote about potential topics of future research with emphasis on the further development of the LB process and conceivable applications and other materials (including specific examples).
  3. To the 33 already existing references we added 10 more, none of them more than 10 years old, to hopefully now better represent the current state of the art.

Thank you again for your great contribution to our paper by doing this review. We hope that we could address all concerns to your satisfaction.

Best regards,

David Schmelz

Reviewer 2 Report

The paper describes the fabrication of antireflection Structures for VIS and NIR on Arbitrarily Shaped Fused Silica Substrates by Colloidal Polystyrene Nanosphere Lithography. The approach is simple with the possibility for upscaling. However, the paper still lacks important details to improve its quality. Some comments and suggestions are shown below:

1.       Toward the up-scaling and reproducibility, and since a limited scale in SEM is given, what is the largest surface area that can be covered with a monolayer PS mask?

2.       The optimization of O2 plasma treatment toward PS shrinkage should be shown, such as the effect of time (or power) vs particle size reduction and was the thin connection line between spheres after plasma treatment occurred at some point? How reproducible is this step?

3.       The O2-RIE and ICP-RIE procedure in plano-convex aspherical lenses needs to be explained in detail.

4.       In regards to Figure 6, the explanation about the structure sample and structure surface are not clear, as well as the caption in the figure.

5.       The causes of the gap between Tspec and Ttotal in shorter wavelengths were justified as the results of the pattern defect. This could be discussed in depth with supporting relevant references or even a simulation, such as, through the significance of the conical surface in creating AR effects.

6.       Related to Figure 7, why the ref dsp 2” wafer and Tspec structure 2” wafer in both attempts displayed as Figure 7a and b look exactly similar, while the setup in both attempts are different. Are they the same dataset?

A thorough proof reading required.

Author Response

Dear Sir or Madam,

We are very thankful for your thorough review. Your detailed reading was clearly obvious. I am very pleased you did the review because I think with your help, we could improve it a lot. We shortened the abstract, and improved the introduction and the conclusion of our paper. We also added several references. The list with comments, you gave us, was very helpful and following this list, I would like to present what we have improved.

  1. To first answer your question, we have already achieved depositions on 3″-wafers. However, it was done with 600 nm PS nanospheres on Si wafers. For this paper, we used 2″ SiO2 wafers and thought it would be the best to give the reader an impression by showing a photograph. Hence, there is a new figure 1, now showing two images of a patterned sample wafer and a dsp reference wafer. The focus of the camera was once set on the samples and once on the reflected objects.
  2. We present now the results of our O2 plasma optimization (see Figure 5). It is not the most comprehensive study, but it shows the main point which is a linear dependence between the size of the shrunk PS spheres and the etching time. The process is, in principle, reproducible but only on samples with the same dimensions and hence must be adapted when changing the utilized substrate. We were also able to observe the thin connection lines as a result of the van-der-Waals-connection. However, we observed it only in pre-studies with 600 nm PS spheres. For 200 nm spheres, we could observe a slightly thicker connection line at self-assembled PS patterns, but before the treatment with O2 plasma. Hence, we did not include it in the manuscript.
  3. Regarding O2-RIE, Cl-RIE and ICP-RIE (with CHF3), we have added a lot of information to go further into detail. We also mention how we changed the process steps 2 and 4 for the lens sample in comparison to the wafer sample.
  4. Figure 6 (now Figure 8) shows the transmission measurements of the single-side-patterned sample on the left (a) and the transmission of only the patterned surface calculated from the sample transmission. This was actually intended for an easier and faster assessment of the AR effect (how far reaching to an ideal AR effect). However, I understand, that it was not sufficiently explained. Therefore, we added further words for clarification.
  5. We have now discussed the transmission results of Tspec and Ttotal in more depth. We included half a page of discussion for that and also use additional references where we cannot go more into detail. We explain the transmission maximum, the increasing gap between Tspec and Ttotal and also, why both values are decreasing towards shorter wavelengths. Concerning the suggested simulation, we have to admit that 9 days were not enough time to set up a completely new simulation study with enough conscientiousness. However, we hope that the added explanations are sufficient for a full understanding of the presented results.
  6. This was a very good point. Thank you for noticing it. In the original manuscript, the wafer values were taken from Figure 6 (now Figure 8) to serve only as a reference for an estimation of the later measurements. However, thanks to your comment, I got aware that we could instead include the measurements from the wafer samples that we achieved in the different setups. Hence, we did so. Now, there is three diagrams with different results from different measurement setups.

We corrected several mistakes, and hope, there is not more to find.

Thank you again for your careful reading and the valuable ideas and comments, you gave us for improving our manuscript. We hope that we could address all concerns to your satisfaction.

Best regards,

David Schmelz

Round 2

Reviewer 2 Report

The authors have carefully addressed all my concerns. The paper is now ready for publication.

Proof-reading required.